# Opportunities and Challenges in Interpreting and Sharing Personal Genomes

**DOI:** 10.3390/genes10090643

**Published:** 2019-08-25

**Authors:** Irit R. Rubin, Gustavo Glusman

**Affiliations:** Institute for Systems Biology, 401 Terry Ave N, Seattle, WA 98109, USA

**Keywords:** genome sequencing, medical applications, privacy, data sharing, algorithms

## Abstract

The 2019 “Personal Genomes: Accessing, Sharing and Interpretation” conference (Hinxton, UK, 11–12 April 2019) brought together geneticists, bioinformaticians, clinicians and ethicists to promote openness and ethical sharing of personal genome data while protecting the privacy of individuals. The talks at the conference focused on two main topic areas: (1) Technologies and Applications, with emphasis on personal genomics in the context of healthcare. The issues discussed ranged from new technologies impacting and enabling the field, to the interpretation of personal genomes and their integration with other data types. There was particular emphasis and wide discussion on the use of polygenic risk scores to inform precision medicine. (2) Ethical, Legal, and Social Implications, with emphasis on genetic privacy: How to maintain it, how much privacy is possible, and how much privacy do people want? Talks covered the full range of genomic data visibility, from open access to tight control, and diverse aspects of balancing benefits and risks, data ownership, working with individuals and with populations, and promoting citizen science. Both topic areas were illustrated and informed by reports from a wide variety of ongoing projects, which highlighted the need to diversify global databases by increasing representation of understudied populations.

## 1. Introduction

As genetic testing expands out of the research laboratory into medical practice and the direct-to-consumer market, there is increasing public interest in efficient and private analysis of personal genomic variation to inform personal healthcare. There is significant value in wide sharing of genetic data for prioritizing and assessing rare or novel variants and combinations of variants, and for expanding our knowledge of disease risks associated with common variants. On the other hand, personal genetic information is recognized as sensitive for multiple social reasons, raising concerns about privacy and questions about best practices for governance of personal genomics data access.

To address these issues, 130 delegates from 21 countries participated in the 2019 conference on “Personal Genomes: Accessing, Sharing and Interpretation” (PersGen19), which took place at the Wellcome Genome Campus in Hinxton, UK, 11–12 April 2019. The meeting was organized by Drs. Manuel Corpas (Cambridge Precision Medicine, UK), Mahsa Shabani (University of Leuven, Belgium), Mad Price Ball (Open Humans Foundation, USA) and Stephan Beck (University College London, UK). Its main goal was to promote openness and ethical sharing of personal genome data while protecting the privacy of individuals. The conference, bookended by two keynote talks, included six sessions covering diverse aspects of genetic testing, interpretation, data sharing, citizen science, the return of results to research participants, societal challenges of data protection and sharing, and clinical perspectives on genomic data integration into healthcare. In addition to 25 expert talks, the conference included two panel discussions, 21 posters, and a discussion session open to the general public.

For ease of presentation, we chose to organize the talks into two main topic areas (Figure 1): (1) Technologies and Applications, with an emphasis on personal genomics in the context of healthcare, and (2) Ethical, Legal, and Social Implications, with emphasis on genetic privacy. We acknowledge that at times this subdivision was not trivial: As expected for this complex and evolving field, all presenters had to address and balance both the technical and ethical aspects of personal genomics. 

## 2. Topic Area: Technologies and Applications

In the opening keynote lecture, George Church (Harvard University, USA) reported on the achievements of the Personal Genome Project (PGP) [1] and its offshoots. Cell lines, including those generated from PGP participant samples, allow the study of the effects of variants in any cell state that can be recapitulated in vitro. These cells can also be edited using CRISPR and similar technologies. CRISPR editing at a high scale can eliminate entire classes of viral integration sites, resulting in retrovirus-resistant cell lines. Church emphasized that to design such edits, one must use the individual’s genome as a reference, as using the standard human reference results in 37% off-target edits. Gene editing can also be used to introduce a variant of unknown significance that is found in a person’s genome into normal iPS cells in order to evaluate the effect of the variant as the cells are differentiated in various contexts (by means of transcriptome editing—activating transcription factors to get the desired cell state), including being used in organoid systems. With the right assay, one can even identify early stages in the development of processes related to late-onset conditions such as Alzheimer’s Disease. Church’s company, Nebula Genomics, allows customers to receive information based from the sequencing of their entire genome at very little cost, promising full privacy. The genomes are queried and analyzed through homomorphic encryption, but cannot be read directly. The cost is to be paid by researchers who access the encrypted data. Finally, Church predicted that the ‘killer app’ that will popularize personal genomics will be a method for match-making between people who want to avoid having offspring who would suffer from recessively inherited genetic conditions.

Many of the talks in this Topic Area addressed the opportunities, challenges and insights in the integration of personal genomics in healthcare, whether in helping diagnose existing conditions or assessing future risk and identifying candidates for preventive care. This work is mainly being carried out in the context of national genome projects, bringing a valuable perspective on the current lack of diversity in available genetic data. Andres Metspalu (The Estonian Genome Center, Institute of Genomics, University of Tartu, Estonia) reported that the Estonian Biobank already has health and genetics data for 15% of the Estonian population, and that the cost of recruitment and genotyping is already down to 50 Euro per individual. Genotyping allows identifying at-risk people who can benefit from early treatment (e.g., early statin treatment for familial hypercholesterolemia), predicting risk for common, major diseases based on polygenic risk scores (PRS) and lifestyle, and predicting drug response. Naveed Aziz (CGEn—Canada’s National Platform for Genome Sequencing and Analysis) reported findings from whole-genome sequencing of the first cohort of PGP Canada. His main finding was the high cumulative frequency of rare variants: A total of 95% of the sequenced individuals were carriers of at least one recessive protein-altering variant, 25% of which are disease-associated, with about three such variants per person, on average. In his words, “we are all mutants!”. However, not all genome projects focus on disease. Sungwon Jeon (KOGIC, Ulsan National Institute of Science and Technology, Republic of Korea) reported on the Korean Personal Genome Project, with the goal to sequence 100,000 Korean genomes to promote healthy, happy aging. 

New nation-specific personal genome projects are being established to serve the needs of underrepresented populations. Anuradha Acharya (MapMyGenome India, Ltd.) addressed the difficulty generalizing European-centric knowledge to understudied populations and the need for expanded knowledge of population-specific risk variants, biomarkers, and drug responses. Lorenza Haddad Talancon (Codigo 46, Mexico) further stressed how risk calculation and the advice given to participants must take into consideration lifestyle differences and limited access to healthcare, especially preventive care. Finally, and in contrast with past experience where marginalized populations were treated merely as research subjects, Genomics Aotearoa, presented by Ben Te Aika (University of Otago and University of Auckland, New Zealand) is a genomic science data repository and research platform that respects the sovereignty of the Maori people. The program centers Maori researchers and works closely with the Maori community.

A recurring area of interest was the computation, validation and interpretation of polygenic risk scores (PRS). PRS have clear advantages over the interpretation of individual genome variants, but several caveats to their use were identified, including issues with their definition, accuracy, and applicability to understudied populations [2], and predictive power [3]. Heidi Marjonen (National Institute for Health and Welfare, Helsinki, Finland) discussed predicting risk for common diseases based on polygenic risk scores compared to traditional risk factors. In their study, high polygenic risk scores were associated with earlier onset. Cathryn Lewis (King’s College, London, UK) pointed out that many third-party apps report results from single SNPs for conditions that are highly polygenic. She proposed a tool for imputing polygenic risk from genotyping data and communicating that risk to users on a population-level distribution and within medical context [4]. Gustavo Glusman (Institute for Systems Biology, Seattle, USA) discussed various possible interpretation contexts (the individual, their family, a designed population, and a real-world population) and the utility of fast genome [5] and electronic health records [6] comparison via fingerprinting algorithms, including a method for modeling polygenic risk scores to enhance their generality across populations. Vincent Plagnol (GenomicsPlc, Oxford, UK) further showed how the prediction of genetic risk for common, polygenic conditions can be improved by leveraging correlations between related traits [7].

Once data about variants and risk is obtained, are they being used in ways that are helpful to the people who donated the samples? Reecha Sofat (University College, London, UK) presented AboutMe, a platform for accessing genomic data in order to inform routine healthcare decisions. Nicki Taverner (Cardiff University and All Wales Medical Genetics Service, UK) raised questions regarding the impact of genomic testing, considering that it relies on many variants with unknown significance. The results may reveal incidental findings people are not prepared for, such as late-onset conditions. Can the tested people be offered management strategies that are effective or is the effect just increased anxiety over future health (their own and that of potential offspring)? [8] Saskia Sanderson (University College London, UK) reported that genomic prediction of disease risk alone has little effect on behaviors such as diet and exercise, but has a strong effect on medical behavior including sharing information with doctors or following through with medications. Furthermore, they observed that mental distress following the prediction of genetic risk is usually small and temporary [9]. Frances Elmslie (St George’s, University of London, UK) gave examples showing how genomic testing can assist diagnosis when symptoms are very general and can have many different causes (e.g., severe intellectual disability), as long as genomic findings are properly filtered compared to population and family members. In his view, the consent process should include the expectation of incidental findings.

## 3. Topic Area: Ethical, Legal, and Social Implications

This Topic Area included discussion of multiple axes of ethical, legal and social aspects of genetic data, including ownership, sharing, privacy and regulation. Talks covered the full range of genomic data visibility, from fully open access in the public domain to very tightly controlled in research context. Joanne Hackett (Genomics England, UK) presented on the 100,000 Genomes Project, including genomes related to rare diseases and to cancer. Of the rare disease genomes, 20%–25% have actionable findings, and about 50% of the cancer genomes have findings with clinical potential. Sharing genomic and clinical data with researchers has resulted in finding people who may have a condition of interest from among previously undiagnosed patients. To enable such interactions while limiting harm to patients, researchers with approved projects have access to de-identified subsets of the data within a monitored secure environment. Hackett also announced Genomics England’s intent to sequence five million genomes in five years [10], and stressed that “the individual owns their data; we are just custodians”. Ciara Staunton (School of Law, Middlesex University London, UK and Centre for Biomedicine, EURAC, Italy) related how, following a history of exploitative research, South Africa passed the Protection of Personal Information Act (POPIA, which will come into force in 2020) to protect the right to privacy in South Africa. However, in contrast with similar EU laws, POPIA lacks provisions for using data for research [11]. Recently legal, ethical, and scientific experts have convened to identify issues that need addressing in order to allow the development of a code of conduct that will foster the sharing of genomic data under proper oversight. Gary Saunders (ELIXIR, Hinxton, UK) presented the ELIXIR federated network of human data resources across member countries in Europe, the goal of which is to allow researchers across the continent efficient access to large datasets for secondary research, aiming to share one million genomes transnationally by 2022 [12]. This will require defining genomic and clinical information standards, developing common application programming interfaces, establishing a repository of tools and services, and working through secure, federated cloud environments. Pascal Borry (Centre for Biomedical Ethics and Law, Department of Public Health and Primary Care, KU Leuven, Belgium) discussed how customers of direct-to-consumer (DTC) genomic services first have their genomic data interpreted outside of the clinical context, and then come to clinicians asking for interpretation of data obtained privately and potentially also become participants in research conducted by private companies. This raises challenges with regard to the privacy of raw data and interpreted data of the consumers and their family members. Borry stressed that individuals are typically willing to share data for research, and that the platforms enabling such sharing need to be subject to ongoing scrutiny in order to maintain an environment of trust and altruism.

Citizen science and participation in healthcare and research were also discussed in various ways—along with their intersection with law enforcement. Colin Smith (University of Brighton, UK) described the Patients Know Best platform [13], which empowers patients to manage their care through patient-controlled electronic health records. Further aiming to enable deep phenotyping, Monica Munoz-Torres (Oregon State University, Corvallis, Oregon, USA) described a way to involve patients in the diagnosis process by using patient-centered tools that translate Human Phenotype Ontology (HPO) terms to lay language [14]. Bastian Greshake Tzovaras (Lawrence Berkeley National Laboratory, Berkeley, USA and Open Humans Foundation, USA) described openSNP [15], a crowdsourced repository for genotyping data that allows individuals to deposit their genotypes and phenotype data in the public domain, and presented results from an analysis of the demographics and motivations of people who openly share genomic data [16]. Maurice Gleeson (International Society of Genetic Genealogy) [17] discussed the increasing interest in using DTC DNA tests for the purpose of studying ancestry. Members of the public have run thousands of projects based on Y chromosome haplotyping, mitochondrial DNA haplotyping, and more recently also based on autosomal DNA. There is interest in following the geographic spread of lineages, comparing genomic information to ancient genealogical records, admixture information, and identifying unknown persons or their remains. Finally, Christi Guerrini (Baylor College of Medicine, Center for Medical Ethics and Health Policy, Houston, Texas, USA) discussed how forensic material can be matched to genealogy databases to identify close relatives of suspects, and SNP searches can identify more and more distant relatives, with a concomitant increase in the risk of false positives. Guerrini reported that at the moment the public is supportive of such use of genealogical data for the purpose of solving serious crimes, citing ~85% support for such use in the case of violent crime, and ~46% for nonviolent crime [18]. DTC testing companies vary in how permissive they are of letting law enforcement use their databases.

The closing keynote talk was given by Yaniv Erlich (MyHeritage, Israel), and covered many aspects of privacy of genomic data. Erlich described how current tools already allow reidentification of individuals based on haplotype data in many cases, and warned that the success rate will increase as more personal genomes become accessible [19]. It is also easy to use haplotype information to impute unpublished genetic risk information. Given that, Erlich recommends against making unrealistic promises of privacy to study participants. Instead, he proposes systems based on creating trust between research participants and researchers by clearly defining how the data can be used, reporting data use to the participants, creating an auditing mechanism that follows data usage and flags suspicious use, creating a reputation system for researchers and participants, and implementing a dynamic model for consent whereby a participant can opt in or out of additional studies over time [20].

## 4. Algorithms

While the conference had no explicit focus on algorithms, these were central to several talks, even if not always explicitly. Indeed, it became apparent that the growing computational ability to share, access, compare and interpret personal genomic data is instrumental for realizing the translational opportunities of genomics. Furthermore, algorithms can significantly affect many aspects of societal impact, both positively through democratization and facilitation of data sharing, and sometimes negatively, by increasing the risk of loss of privacy. As described above, several talks discussed aspects of genome interpretation and disease risk prediction through polygenic risk scoring. Other algorithms of relevance to personal genomics included Mendelian randomization to improve drug discovery, algorithms supporting the logistics of large-scale genomic data sharing and permissioning, algorithms for probabilistic integration of data types to enable reidentification, and homomorphic encryption and genome fingerprints for (respectively) querying and comparing genome data, without having full access.

With the fast-growing number of individuals accessing their genetic data worldwide, the 2019 conference on “Personal Genomes: Accessing, Sharing and Interpretation” was very timely and succeeded at framing the many opportunities and challenges in interpreting and integrating such data into healthcare systems in both effective and ethical ways. We look forward to future follow-up conferences to chart and track the progress of this fast-growing field.

## Figures and Tables

**Figure 1 genes-10-00643-f001:**
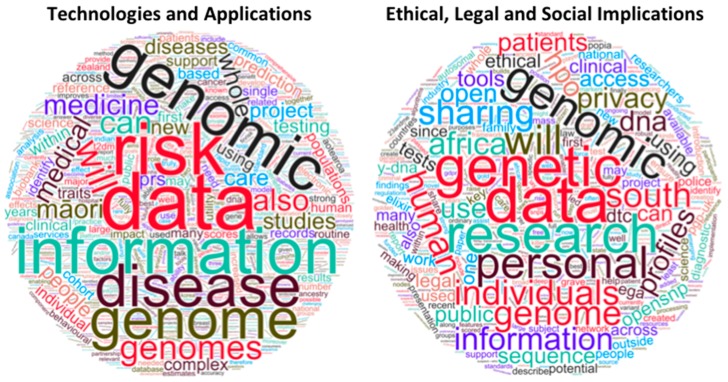
Word clouds from the titles and abstracts of the two Topic Areas: Technologies and Applications (**left**) and Ethical, Legal and Social Implications (**right**). Font size reflects word frequency; colors and word directions are arbitrary. Generated using WordClouds.com.

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
