# Peer review of "Opportunities and Challenges in Interpreting and Sharing Personal Genomes"

_genes, 2019, doi:10.3390/genes10090643_

Round 1

Reviewer 1 Report

Line 9 & Line 34:
"The first “Personal Genomes: Accessing, Sharing and Interpretation” conference…"

The word "first" implies a lot of certainty regarding future conferences. I think it's clearer and less prone to potential error to simply specify the year of this conference, e.g. "The 2019 “Personal Genomes: Accessing, Sharing and Interpretation” conference…"

Line 44:
"Lively" is a value judgement. I think I expect a conference report to be more neutral, maybe instead this should describe the number of panels?

Line 59: "Cell lines from PGP participants allow the study of the effects of variants in any cell state…"

This isn't unique to PGP participants: cell lines from any source can be used in this manner. This report should be careful about distinguishing between what the PGP provenance brings to cell lines vs. generally report on recent advances in biotechnology.

Line 121: "the utility of fast genome and electronic health records comparison via fingerprints"
I think this sentence would be clearer if you use the phrase "fingerprinting algorithms" instead of "fingerprints".

Line 226: "Surprisingly, there was very little mention of artificial intelligence…"

It's not so surprising to me... (nor is, thankfully, the absence of any mention of blockchain).

I'd recommend dropping this statement. I think it's a value judgement that depends a lot on ones understanding and expectations for the appropriateness of AI.

For the record, why I would call it "unsurprising": Machine learning and artificial intelligence are focused on optimization, they aren't intended to advance understanding of underlying mechanisms. This is especially true for more recent work like "deep learning", which has an inherent lack of explainability. (Indeed, lack of explainability makes AI inherently untrustworthy and dangerous to deploy in scenarios where the consequences of algorithmic errors are dangerous – i.e. rather than "less efficiency", an increased risk of severe outcomes like medical errors.) Genomics research is broadly interested in understanding of the underlying biological mechanisms. We should expect that understanding to be more robustly helpful for downstream translational research, and expanding general knowledge is an intrinsic goal of science. When the goal is "understanding", broadly speaking, ML/AI aren't very useful.

Author Response

Line 9 & Line 34: The word "first" implies a lot of certainty regarding future conferences. I think it's clearer and less prone to potential error to simply specify the year of this conference, e.g. "The 2019 “Personal Genomes: Accessing, Sharing and Interpretation” conference…” Response: We replaced the word ‘first’ with ‘2019’ as recommended. We similarly corrected a third instance on line 229. Line 44: "Lively" is a value judgement. I think I expect a conference report to be more neutral, maybe instead this should describe the number of panels? Response: We replaced the word ‘lively’ with ’two’, as suggested. Line 59: "Cell lines from PGP participants allow the study of the effects of variants in any cell state…” This isn't unique to PGP participants: cell lines from any source can be used in this manner. This report should be careful about distinguishing between what the PGP provenance brings to cell lines vs. generally report on recent advances in biotechnology. Response: We modified this sentence to disambiguate the intent:"Cell lines, including those generated from PGP participant samples, allow the study…" Line 121: "the utility of fast genome and electronic health records comparison via fingerprints” I think this sentence would be clearer if you use the phrase "fingerprinting algorithms" instead of "fingerprints”. Response: We edited the sentence as recommended. Line 226: "Surprisingly, there was very little mention of artificial intelligence…” It's not so surprising to me... (nor is, thankfully, the absence of any mention of blockchain). I'd recommend dropping this statement. I think it's a value judgement that depends a lot on ones understanding and expectations for the appropriateness of AI. Response: We dropped this sentence as recommended.

Reviewer 2 Report

This report is a fair and thoughtful overview of the first Personal Genomes conference. The organization of the presentations by two topic areas is helpful, and the presentations are nicely summarized. It's also helpful that the authors highlighted the presenters' emphasis on polygenic risk scores.

Author Response

We thank the reviewer for the positive appraisal.